# Training Tobacco Treatment Specialists through Virtual Asynchronous Learning

**DOI:** 10.3390/ijerph19063201

**Published:** 2022-03-09

**Authors:** Audrey Darville, Kathy Rademacher, Amanda T. Wiggins, Mary Grace Lenhof, Ellen J. Hahn

**Affiliations:** College of Nursing, University of Kentucky, Lexington, KY 40504, USA; kathy.rademacher@uky.edu (K.R.); atwiggins@uky.edu (A.T.W.); mary.lenhof@uky.edu (M.G.L.); ejhahn00@email.uky.edu (E.J.H.)

**Keywords:** tobacco dependence treatment, health professional training, evidence-based practice, tobacco cessation, online training

## Abstract

Tobacco dependence is a prevalent, chronic, and complex addiction that often leads to long-term disease and death. However, few healthcare providers are sufficiently trained and feel comfortable in delivering tobacco dependence treatment. The purpose of the study was to examine the effectiveness of an accredited online Tobacco Treatment Specialist (TTS) training program that uses a novel, asynchronous approach. We compared the characteristics of participants who completed the program to those who did not complete the program. Changes in knowledge and attitudes in providing tobacco dependence treatment were measured, and satisfaction with the program and intent to pursue national certification were assessed. Participants who were more likely to complete the program were those who discussed quitting less frequently with patients prior to course enrollment. These participants had a significant increase in knowledge and high satisfaction with the course. Approximately half of participants who completed the program indicated that they would pursue obtaining a national certificate in tobacco dependence treatment in the next 2 years.

## 1. Introduction

Tobacco dependence is a complex, chronic, relapsing addiction to nicotine that significantly contributes to chronic disease development and death [1]. Yet most clinicians do not have advanced or specialized training in evidence-based treatments for tobacco dependence [2], and they express discomfort with counseling patients about cessation [3]. There are clear benefits of intensive tobacco treatment for persons with mental illness [4], cancer [5], cardiovascular disease [6], and women during pregnancy [7]. 

The World Health Organization (WHO) has established a framework to reverse the “tobacco epidemic” globally [8], identifying a need for comprehensive guidelines for treating tobacco use. Evidence-based, intensive tobacco treatment enhances quitting, particularly when provided by persons with advanced training in these treatment strategies [9,10]. 

Yet there is a lack of adequate knowledge and skills-based training for healthcare providers in tobacco dependence treatment, including those needed to deliver intensive counseling. Notably, it has only been in recent years that content, albeit sparse, related to treating tobacco dependence has been incorporated into medical school curricula in the United States [11], despite a globally identified need for this training [12,13].

The “5 As Model” (Ask, Advise, Assess, Assist, and Arrange) has been a moderately effective framework to guide providers in treating tobacco dependence [14]. Despite adoption of the 5 As model by the U.S. Preventive Services Task Force, providers typically ask patients about tobacco use and advise them to quit, but only about half provide assistance, and relatively few offer follow-up after treatment is initiated [15]. 

Evidence supports the need for targeted and tailored treatment approaches for vulnerable populations [16,17]. A changing landscape of tobacco treatment delivery is also emerging, including mHealth, text to quit, and telehealth [18]. Recent advances in lung cancer screening provide an opportunity to treat tobacco use on a large scale with high-risk populations [19,20].

The Association for Treating Tobacco Use and Dependence (ATTUD), an international professional organization for Tobacco Treatment Specialists, formalized national competency standards for tobacco treatment specialists (TTSs) in 2004 [21]. Tobacco treatment specialists acquire unique training to deliver interventions across multiple modalities and intensities, with learners developing skills in motivational interviewing and counseling [22]. This specialized training ensures use of the best evidence currently available for providing tobacco treatment. Tobacco treatment specialists are now trained and recognized internationally as treatment providers.

The Council for Tobacco Treatment Training Programs (CTTTP), as an accrediting organization, ensures TTS training programs meet the ATTUD standards. Currently, there are 22 accredited programs internationally [23] including the fully online BREATHE (Bridging Research Efforts and Advocacy Toward Healthy Environments) Tobacco Treatment Specialist Training Program. Upon successful completion of an accredited TTS training program, participants are eligible to apply for a standardized exam to receive a National Certificate in Tobacco Treatment Practice (NCTTP). Over 9000 TTSs have been trained in accredited programs from 2016 through 2020 [24]. 

### 1.1. Description of the BREATHE Accredited TTS Training Program

BREATHE faculty and their public health, academic, and community partners identified a need in Kentucky for a tobacco treatment training that would be accessible to providers in rural and remote settings. We developed the BREATHE Tobacco Treatment Specialist (TTS) Training as a novel online-only program, incorporating video technologies delivered in an asynchronous format, thus, allowing for convenience of access. 

The asynchronous format provided an accessible online option for international participants interested in training. Faculty experts from the University of Kentucky (UK), Western Kentucky University and persons involved in tobacco treatment and prevention in the public health sector used the ATTUD competency standards to structure the training.

We developed and evaluated an online-only, asynchronous, self-paced format that incorporates evidence-based online teaching strategies, including video-based simulations. The training concludes with a simulated patient practice case, scheduled as a one-hour, real-time interaction with a course instructor. We developed the training using the Canvas© online-learning platform, following the guidance of instructional design and online learning experts at our institution. We pilot tested the training prior to applying for accreditation and used participant feedback as the basis for improving the user experience and developing program process and outcome evaluation procedures. 

The training is based on the ATTUD core competencies foundational to TTS training curricula [22]. The competencies guide the development of knowledge and skills needed to move from awareness to knowledge and ultimate proficiency in the delivery of tobacco treatment services across eleven domains: (1) Tobacco Dependence Knowledge and Education, (2) Counseling Skills, (3) Assessment Interview, (4) Treatment Planning, (5) Pharmacotherapy, (6) Relapse Prevention, (7) Diversity and Specific Health Issues, (8) Documentation and Evaluation, (9) Professional Resources, (10) Law and Ethics, and (11) Professional Development. 

Our program organizes these domains into five modules, each taking approximately five hours to complete. Each module includes activities designed to engage learners: didactic content, video demonstrations, written assignments with individualized feedback, quizzes, and supplemental resources that can be tailored to the participant’s needs. 

Successful completion is based on written assignments and achieving a grade of at least 75% on a comprehensive written case, module-specific quizzes, and a pharmacology and final exam. The total training time is a minimum of 27 contact hours, which are logged in Canvas©. Participants are provided with an overview of the expected course progression at the start of the training (Table 1). The course is self-paced, and it is advised to maintain steady progress. It is expected that the entire training be completed within 6 months.

### 1.2. Purpose and Objectives

The purpose of this paper is to evaluate the effectiveness of the BREATHE TTS Training Program. We compared the characteristics of those who completed and did not complete the training, and we assessed the changes in knowledge and attitudes after completion of the TTS training, evaluated satisfaction with the training and format, and intent to pursue national certification.

The goal of this evaluation study was to examine the association between learner characteristics and successful program completion, evaluate time to completion, changes in knowledge acquisition, satisfaction with the training, and subsequent interest in certification. 

## 2. Materials and Methods

### 2.1. Study Design

The evaluation study was a non-randomized prospective design. Registration, pre- and post-training knowledge acquisition scores, and program completion data for all participants were collected regardless of participation in the research study. If participants provided consent to take part in the research study, registration and program completion data were linked to the post-training survey data, collected via online surveys at three time points: 1-, 3-, and 6-months post-training. Participants were given an opportunity to opt out of subsequent post-training surveys. The research was approved by the university’s medical institutional review board.

### 2.2. Study Population and Sample

The study population were health care workers, healthcare students, and community health personnel caring for persons with tobacco dependence and who participated in the BREATHE TTS Training Program. The training was available to persons who work with tobacco users in a variety of clinical settings (e.g., inpatient, medical and dental ambulatory care, patient navigation, behavioral health counseling, health departments, and other community-based and social services). 

The minimum requirements for the training were based on established criteria for the NCTTP, which are a combination of educational attainment and recent work experience in a human services position (e.g., for HS diploma, 2 years of experience required; none for bachelor’s degree in healthcare/counseling affiliated field). Computer access with camera/microphone capability was also required. 

Participants enrolled in training cohorts occurred every other month starting in September 2017 through December 2019. A total of 210 participants enrolled and completed at least one module of the training during this time span. Participants were located throughout the United States and internationally, with 31% living and/or working in Kentucky. Participants were expected to complete training within a 6-month time frame, though the asynchronous nature of the format allowed for flexibility in completion time. 

### 2.3. Methods and Measures

All surveys were developed using the Qualtrics© software platform. The registration survey included basic demographics (e.g., gender, race/ethnicity, location, and educational level), work characteristics (e.g., role and work setting and prior experience helping people quit tobacco); and additional minimum data set items (e.g., intent to pursue certification) required by the CTTTP. 

Persons completing the training were invited to participate in a post-program 11-item survey to assess the long-term effectiveness and applicability of the training, which required about 15 min to complete. Participants were asked to rate the helpfulness of the training related to four specific skills (motivational counseling, assessing tobacco use and dependence, knowledge of medications used to treat tobacco dependence, and ability to deliver evidence-based treatment), using a 5-item Likert scale ranging from ‘Not at all’ (1) to ‘Very helpful’ (5). 

A mean score was calculated to represent overall helpfulness of the skill development training. Usefulness of the training in their work setting was a single item with responses ranging from ‘Not at all useful’ (1) to ‘Very useful’ (5). A similar scale was used to assess both satisfaction with support materials and ability to effectively provide tobacco treatment with options ranging from ‘Not at all satisfied’ (1) to ’Very satisfied’ (5).’ Participants were asked if they planned to pursue national certification in tobacco treatment within the next two years (yes/no/undecided). 

Participants received the post-program survey at three time points: 1-, 3-, and 6-months post-training completion, with an automatic email reminder each time. The post-program data were linked with data from the registration survey for the evaluation reported here. Survey responses were identifiable by email address, and the date of each response was recorded. 

Program engagement was monitored using the Canvas© online learning platform. Participants completed a pre- and post-program 20-item ‘true-false’ knowledge acquisition quiz to measure their basic knowledge of tobacco control and treatment concepts covered in the training. Possible scores ranged from 0 to 20, with higher scores reflecting greater knowledge acquisition.

### 2.4. Data Analysis

Frequency distributions summarized demographic and workplace characteristics. The Fisher’s exact and Mann–Whitney U test were used to determine unadjusted associations among demographic and workplace characteristics and program completion. The paired samples t-test was used to examine knowledge acquisition using the pre- and post-quiz total scores. Predictors of intention to certify as a tobacco treatment specialist were determined using logistic regression. The Hosmer–Lemeshow test was used to evaluate model fit. All data analysis was conducted using SAS, version 9.4 with an alpha of 0.05. Data were analyzed for participants enrolled prior to January 2020 as COVID-19 impacted participant progression during 2020.

## 3. Results

A total of 210 people enrolled in bi-monthly cohorts of the BREATHE training program between September 2017 and December 2019. Of these, 13 participants provided incomplete demographic data and were removed from the analysis, leaving an effective sample size of 197. Descriptive demographic data are in Table 2 for those who completed the training prior to December 2019 (*n* = 181) and those who did not complete it (*n* = 16). The majority of participants were female (86%) and identified their race/ethnicity as White, non-Hispanic (86%). 

Over one-third (39%) of enrollees had a Bachelor’s degree, and less than half (45%) held a Master’s degree or above. The majority worked in direct patient care (71%), and almost half (48%) discussed quitting tobacco with their patients or others at work on a daily basis. Almost all (92%) of those who started the program completed it. The only participant characteristic associated with completion status was the frequency of discussing quitting tobacco with their patients; those who completed the course discussed quitting less frequently with patients than those who did not complete the course (*p* = 0.02). For those who completed the training, on average, the time to completion was 111.8 days (*SD* = 80.1).

Based on a potential score range of 0–20, the mean pre-knowledge score was 14.5 (*SD* = 2.1) and mean post-knowledge score was 18.2 (*SD* = 2.0), resulting in an average increase in mean overall knowledge of almost 4 points (M change = 3.7, *SD* = 2.6; *p* < 0.001).

Among those with post-training survey data from 137 program completers, the mean scores for helpfulness (4.61, *SD* = 0.45, see Table 3) and usefulness of the training (4.69, *SD* = 0.61) were high on a 5-point scale. The mean scores for satisfaction with the training materials (4.47, *SD* = 0.60) and the ability to effectively provide tobacco treatment (4.42, *SD* = 0.59) were also rated high. There were no differences in attitudes over time, indicating that these scores remained stable from 1 to 6 months post-training.

Over half (54%) reported the intention to pursue a national certificate in tobacco dependence treatment in the next 2 years; age, race/ethnicity, education, workplace setting, frequency of discussing tobacco treatment in the workplace, helpfulness and usefulness of the training, satisfaction with training materials, and ability to provide tobacco treatment post-training were not significant predictors of intent to pursue certification.

## 4. Discussion

We developed and evaluated the BREATHE Online TTS Training Program in response to a community need for access to online, asynchronous provider training for tobacco treatment. We explored associations between learner characteristics and program completion, evaluated time to completion, changes in knowledge acquisition, satisfaction with the training, and factors associated with subsequent interest in certification. 

Participants who completed versus did not complete the BREATHE TTS program did not vary by gender, race-ethnicity, or educational level in this sample of participants. However, despite having a broad geographic reach, this sample of training participants were not gender- or racially-ethnically diverse. While TTSs are predominantly female (80%) and non-Hispanic white (70%) [24], our sample was notably less diverse. 

It is critically important that TTS training programs reach historically marginalized, diverse healthcare providers and other community health leaders. We plan to expand our reach and offer TTS scholarship support to provide targeted incentives for training to increase racial and ethnic diversity in our participants and those working in medically underserved and rural communities in our state. 

While most participants completed the BREATHE Online TTS training program, 8% did not complete the program. Participant attrition may be higher in online programs than in more traditional, in-person settings [25]. While it is encouraging that completers did not differ from non-completers, we aim to better understand the reasons for participant attrition to inform the development of strategies to maximize program completion. Our team meets weekly to review participants’ progression, and we reach out to individuals to offer assistance in reducing barriers to completion. 

Anecdotally, participants face a variety of challenges, including personal and family illnesses, job changes, shifting role responsibilities, and other competing demands. With the flexible time that we allow for training progression, these challenges may have a greater impact on an individual’s ability to remain focused and complete the training compared to a scheduled, synchronous training, which generally takes place over a few days. 

A few participants expressed concern that online learning was more challenging than anticipated, yet the majority of participants expressed appreciation for the flexible format. It is challenging to strike a balance between setting reasonable time-to-completion expectations and maintaining flexibility. While the average time to completion was close to 16 weeks, a few participants remained engaged for extended periods of time (range 15—652 days). Allowing flexibility in time-to-completion can be both a strength and challenge of the virtual, asynchronous training format. 

Demand for healthcare providers with specialized skills in treating tobacco dependence remains high, with over 9000 treatment specialists trained and accredited training programs expanding internationally [24]. There is a growing demand for TTSs with specialized skills in Lung Cancer Screening programs to increase access to high-quality cessation for high risk tobacco users [26]. Participants in our online training program demonstrated significant gains in knowledge and skills. For those who completed the post-training surveys, high levels of satisfaction with the training and resources provided were reported. Participants found the training helpful in developing a wide range of tobacco treatment skills and useful in their practice settings. Notably, these attitudes persisted over time. 

The proportion of participants who reported intent to seek the National Certificate in Tobacco Treatment Practice (NCTTP) was consistent with synchronous, non-virtual programs (54% vs. 52.1%), as reported by accredited training programs overall [24]. Interestingly, in a 2018 analysis of 74 BREATHE TTS training completers, we found that 61% intended to pursue the NCTTP (unpublished raw data). The barriers cited by participants to pursuing the NCTTP in the 2018 analysis were the requirement to complete 240 h of clinical work in tobacco treatment over 2 years and the cost of the certificate exam ($300). Additional data were not available for the sample used in the analysis reported here. 

This evaluation has limitations. First, while all participants in the included cohorts (*N* = 210) were invited to complete post-training surveys, only 65% provided survey data resulting in the likelihood of selection bias. Those who had more positive attitudes toward the training and who exhibited high levels of satisfaction with the program may have been more likely to complete post-training surveys. Second, the limited geographic and demographic diversity of our participants limits the generalizability of the program evaluation findings. 

While no significant differences were found in those completing the training versus not completing based on gender, race/ethnicity, or education, the majority of participants were non-Hispanic women with high education levels, biasing these results and limiting generalization to training programs serving more diverse populations. It should be noted that the analysis is limited to the online format only, as our program is novel in that it has never had an in-person component. Published research on the TTS training method satisfaction is limited, with one prior study comparing a novel Train-the-Trainer program delivery to a standard TTS training program finding no differences in either knowledge acquisition or participant satisfaction [27], with high scores of 3.81 vs. 3.84 for satisfaction on a 4-point scale, similar to our results.

## 5. Conclusions

The BREATHE all-online, asynchronous format for tobacco treatment specialist training demonstrated sustained interest and positive learner outcomes, thus, demonstrating the effectiveness of our novel approach. Having an established online training program prior to the COVID-19 pandemic allowed us to continue to reach participants. While not all learners prefer the virtual learning format, we demonstrated that training can be delivered in an asynchronous, all-online, flexible learning platform, providing knowledge and skill acquisition of accredited content to meet diverse learning needs. 

Having a virtual option for learning can reach clinicians who may have time or travel barriers to attend synchronous or in-person training. Our asynchronous, online program demonstrated positive learner outcomes with high levels of participant completion, knowledge acquisition, satisfaction, and perceived usefulness of the training after completion. 

## Figures and Tables

**Table 1 ijerph-19-03201-t001:** Check list for course completion.

Check List for Course Completion	
Weeks 1–3	Welcome/Add a Picture/Email Preferences
	Pre-Test
	Module 1
	Module 1 Evaluation
	Module 2
	Module 2 Evaluation
Weeks 3–5	Module 3 (including the Assessment and Treatment Assignments)
	Module 3 Evaluation
Weeks 5–7	Module 4 (including the Pharmacology Quiz)
	Module 4 Evaluation
	Module 5 (including the Relapse Case Study and Written Assignment)
	Module 5 Evaluation
Week 8	Post-Test
	Video Simulated Patient (via Zoom)
	TTS Simulation Self Reflection
	Written Case Study
	TTS Program Effectiveness and Satisfaction Survey
	TTS Final Exam
	Receive Completion Certificate

**Table 2 ijerph-19-03201-t002:** Demographics and workplace characteristics by program completion.

	Total Sample (*n*= 197) *n* (%)	Completed (*n* = 181) *n* (%)	Did Not Complete (*n* = 16) *n* (%)	*p*
Gender				0.43 ^a^
Female	156 (86.2%)	146 (86.9%)	10 (76.9%)	
Male	24 (13.3%)	21 (12.5%)	3 (23.1%)	
Other	1 (0.5%)	1 (0.6%)	0 (0.0%)	
Race/Ethnicity				0.99 ^a^
White, non-Hispanic	158 (86.3%)	146 (85.9%)	12 (92.3%)	
Other race/ethnicity (Hispanic, Asian, Black or African American, Alaskan Native, Mixed)	25 (13.7%)	24 (14.1%)	1 (7.7%)	
Education				0.95 ^b^
High school or Associate’s degree	30 (16.1%)	28 (16.5%)	2 (12.5%)	
Bachelor’s degree	72 (38.7%)	65 (38.2%)	7 (443.8%)	
Master’s degree or above	84 (45.2%)	77 (45.3%)	7 (43.8%)	
Workplace setting				0.41 ^a^
Direct patient care	136 (70.8%)	127 (71.7%)	9 (60.0%)	
Public or community health	49 (25.5%)	43 (24.3%)	6 (40.0%)	
Academia	7 (3.7%)	7 (4.0%)	0 (0.0%)	
Frequency discussing quitting tobacco with patients or others				0.012 ^b^
Never or very little	53 (27.7%)	51 (29.1%)	2 (12.5%)	
Often (1–4 times per week)	46 (24.1%)	45 (25.7%)	1 (6.2%)	
Daily	92 (48.2%)	79 (45.1%)	13 (81.3%)	

^a^*p* from Fisher’s exact test; ^b^
*p* from Mann–Whitney U test.

**Table 3 ijerph-19-03201-t003:** Post-training attitudes.

Attribute (*n* = 137)	Mean (*SD*) *
Helpfulness of training in developing/improving motivational counseling, accessing tobacco use and dependence, knowledge of cessation medications and/or delivery of evidence-based treatment in the clinical setting	4.61 (0.45)
Usefulness of the information for your clinical setting	4.69 (0.61)
Satisfaction with support materials	4.47 (0.60)
Satisfaction with your ability to effectively provide tobacco treatment	4.42 (0.59)

* 5-point Likert Scale from Least (1) to Most (5).

## Data Availability

The data presented in this study are available on request from the corresponding author.

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
