# Peer review of "Training Tobacco Treatment Specialists through Virtual Asynchronous Learning"

_ijerph, 2022, doi:10.3390/ijerph19063201_

Round 1

Reviewer 1 Report

This is an interesting study and well-written manuscript evaluating a novel, asynchronous training program for Tobacco Treatment Specialists. TTS training is incredibly important for training up the healthcare workforce in evidence-based treatment for smoking cessation. Most TTS programs have historically been offered in person, but some programs have started to offer virtual trainings. This evaluation of a novel, asynchronous program is timely (especially as we consider whether training programs can be moved to online formats successfully in the current covid pandemic). I appreciate the opportunity to review this manuscript. I have a few suggestions:

1) Do the authors have Satisfaction data from a non-virtual program that they could share (i.e., do they have an in-person TTS training in addition to the virtual option) as a reference for how the attitudes toward the training may differ between a virtual and non-virtual program? Or could they discuss their Satisfaction data in relation to any other published satisfaction data from in-person TTS programs? I know this was not that comparison of interest in this study, but "Satisfaction with support materials" and "Satisfaction with your ability to effectively provide TT" are a little lower than the other items (still high...ie., above 4). But I wonder if this is because the program is offered virtually (i.e., challenges with online learning) or if those satisfaction ratings are similar to other in-person programs. 

2) In the Discussion, paragraph 4 where you describe the demand for healthcare providers with specialized tobacco treatment skills, you could mention a recent article that makes the case for cross training shared decision making providers as TTS in the context of LCS (Roughgarden et al, DOI: 10.1016/j.amepre.2021.04.021). I agree that the demand is there, but the utilization of TTS training is lower than we'd hope to see for certain clinical specialties despite the high risk nature of the patient population. 

Author Response

Thank you for your review. We have made the following edits to address your concerns:

Point 1) Do the authors have Satisfaction data from a non-virtual program that they could share (i.e., do they have an in-person TTS training in addition to the virtual option) as a reference for how the attitudes toward the training may differ between a virtual and non-virtual program? Or could they discuss their Satisfaction data in relation to any other published satisfaction data from in-person TTS programs? I know this was not that comparison of interest in this study, but "Satisfaction with support materials" and "Satisfaction with your ability to effectively provide TT" are a little lower than the other items (still high...ie., above 4). But I wonder if this is because the program is offered virtually (i.e., challenges with online learning) or if those satisfaction ratings are similar to other in-person programs. 

Response 1: Our program has been in an asynchronous, online format since we launched, so there is no comparison data. We have added content in the discussion referencing another program (UMass), which published comparison data regarding 2 different approaches to in-person training. We have not found any published studies on tobacco treatment training programs comparing in-person to virtual format satisfaction. We have added content to the end of the discussion section (lines 281 ff) to provide context for our reported scores.  

Point 2) In the Discussion, paragraph 4 where you describe the demand for healthcare providers with specialized tobacco treatment skills, you could mention a recent article that makes the case for cross training shared decision making providers as TTS in the context of LCS (Roughgarden et al, DOI: 10.1016/j.amepre.2021.04.021). I agree that the demand is there, but the utilization of TTS training is lower than we'd hope to see for certain clinical specialties despite the high risk nature of the patient population. 

Response 2: Thank you for this context/citation. We have added it to the discussion section (line 254 ff). Our training, and that of other accredited programs, highlight the services TTSs can provide in lung cancer screening. 

Reviewer 2 Report

This study assessed the effectiveness of an accredited online TTS training program using a novel and asynchronous approach among tobacco treatment specialists. Overall, the findings from this paper are interesting and important, but a few things need to be considered and improved.

  1. Abstract, the first sentence should be “often leads to”.
  2. Introduction, I would recommend including more citations regarding the tobacco treatment specialists. For example, “Tobacco treatment specialists acquire unique training to deliver interventions across multiple modalities and intensities, with learners developing skills in motivational interviewing and counseling” should have a citation here.
  3. Methods, I am wondering if there are any eligible criteria for the participant (like age, working experience/years in the related fields, etc.).
  4. For table 2, it seems like most of the selected population is non-Hispanic women with higher education levels. Will this bias the results, or the findings be generalizable to the overall population?
  5. Results, page 5 line 194, complete data were available for 233 participants, but the total is 211 in line 175 on the same page. Please clarify.
  6. Discussion, “BREATHE TTS program completion did not vary by race-ethnicity”, I do not agree with this conclusion. Actually, the population is not diverse, and I think this should be included in the limitation part. And you can not say this is consistent with citation 23. That one included a more diverse population though the majority is White.
  7. Authors may need to specify the other race population like Hispanic, Black, etc.

Author Response

Thank you for your thoughtful review. We have made the following edits based on your comments:
Point 1. Abstract, the first sentence should be “often leads to”

Response 1: Wording change made.

Point 2. Introduction, I would recommend including more citations regarding the tobacco treatment specialists. For example, “Tobacco treatment specialists acquire unique training to deliver interventions across multiple modalities and intensities, with learners developing skills in motivational interviewing and counseling” should have a citation here.

Response 2: Citation added: Sheffer, et. al.; Training Issues Committee of the Association for the Treatment of Tobacco Use and Dependence. Increasing the quality and availability of evidence-based treatment for tobacco dependence through unified certification of tobacco treatment specialists. J Smok Cessat 2016, 11(4), 229-235. doi: 10.1017/jsc.2014.30.

Point 3: Methods, I am wondering if there are any eligible criteria for the participant (like age, working experience/years in the related fields, etc.).

Response 3: Thank you for noting this omission in our study population description. Please see lines 130-134.

Point 4: For table 2, it seems like most of the selected population is non-Hispanic women with higher education levels. Will this bias the results, or the findings be generalizable to the overall population?

Response 4: We have added a fuller discussion of potential bias in the limitations section (see lines 275 ff.).

Point 5: Results, page 5 line 194, complete data were available for 233 participants, but the total is 211 in line 175 on the same page. Please clarify.

Response 5: Thank you for noting this discrepancy. We went back and re-ran the analysis as our additional review found that the number of participants during the study time frame were 210. Due to our asynchronous format, some participants were enrolled but did complete the training after the December 2019 period of analysis, and were initially classified as non-completers, which was not accurate (233). We have corrected this and the re-analysis did not result in any changes in our findings. The table and text have been updated. 

Point 6: Discussion, “BREATHE TTS program completion did not vary by race-ethnicity”, I do not agree with this conclusion. Actually, the population is not diverse, and I think this should be included in the limitation part. And you can not say this is consistent with citation 23. That one included a more diverse population though the majority is White.

Response 6: We have added clarification to the opening sentence in the second paragraph of the discussion to reflect that there was no significant variation between program completers and non-completers. We have also noted that our population is less diverse than those trained in CTTTP-accredited programs internationally (former citation 23, now citation 24). We also included this as a limitation (lines 275 ff.).

Point 7: Authors may need to specify the other race population like Hispanic, Black, etc.

Response 7: We have added the participant race/ethnicity categories reported to clarify ‘other’ in the table. As several of these categories had low representation; we did not break this out in the reporting to ensure participant confidentiality. 

Round 2

Reviewer 2 Report

This revised manuscript has been significantly improved. Overall, I am satisfied with their responses and revision. The only comment is that please double check the sample (233, 211 or 210), font consistent (reference 26), and grammar mistakes. Suggest acceptance after improvement.